# Tagging-Assisted Generation Model with Encoder and Decoder Supervision for Aspect Sentiment Triplet Extraction

Xianlong Luo[1,2]    Meng Yang [1,2*]    Yihao Wang[1,2]

[1]School of Computer Science and Engineering, Sun Yat-Sen University
[2]Key Laboratory of Machine Intelligence and Advanced Computing (SYSU),
Ministry of Education, China
luoxlong@mail2.sysu.edu.cn, yangm6@mail.sysu.edu.cn,
wangyh357@mail2.sysu.edu.cn

## Abstract

ASTE (Aspect Sentiment Triplet Extraction) has gained increasing attention. Recent advancements in the ASTE task have been primarily driven by Natural Language Generation-based (NLG) approaches. However, most NLG methods overlook the supervision of the encoder-decoder hidden representations and fail to fully utilize the semantic information provided by the labels to enhance supervision. These limitations can hinder the extraction of implicit aspects and opinions. To address these challenges, we propose a tagging-assisted generation model with encoder and decoder supervision (TAGS), which enhances the supervision of the encoder and decoder through multiple-perspective tagging assistance and label semantic representations. Specifically, TAGS enhances the generation task by integrating an additional sequence tagging task, which improves the encoder's capability to distinguish the words of triplets. Moreover, it utilizes sequence tagging probabilities to guide the decoder, improving the generated content's quality. Furthermore, TAGS employs a self-decoding process for labels to acquire the semantic representations of the labels and aligns the decoder's hidden states with these semantic representations, thereby achieving enhanced semantic supervision for the decoder's hidden states. Extensive experiments on various public benchmarks demonstrate that TAGS achieves state-of-the-art performance.

## 1 Introduction

Aspect Sentiment Triplet Extraction (ASTE) aims to extract sentiment triplets from a sentence, i.e., **A**spect: the aspect term represents an explicit mention of a discussed target, **O**pinion: the mentioned comment terms/phrases, **S**entiment: sentiment polarity of the aspect, holding significant potential in downstream research and applications. Unlike sentence sentiment classification, ASTE emphasizes

the explanation for sentiments, explicitly highlighting the causes of sentiments and the entities to which they are attached. This task involves addressing challenges such as the diversity of emotional expressions and the complexity of linguistic contexts. For instance, in the sentence "Food wise, it's ok but a bit pricey for what you get considering the restaurant isn't a fancy place," three sentiment triplets can be extracted: (food, ok, neutral), (food, pricey, neutral), and (restaurant, isn't a fancy place, neutral).

**Existing Methods** The current mainstream approaches for ASTE can be classified into two categories: sequence tagging-based approaches and sequence generation-based approaches. ASTE employed a sequence tagging method initially introduced by Peng et al. (2020). However, the sequence tagging-based approaches in ASTE fail to capture the semantic information conveyed by the labels, which can result in semantic mismatches in the predicted results (Zhang et al., 2021b). By leveraging the rich label semantic information and mitigating the potential error propagation in pipeline methods (Paolini et al., 2021; Yu et al., 2023), generation methods achieve better performance in ASTE.

Generation-based approaches still face two significant challenges. Firstly, the supervision of hidden representations within encoder-decoder architectures has been overlooked, leading to potential issues such as the degeneration of neural language models and difficulty in identifying distinctive information (Su et al., 2022). In the context of the ASTE task, this oversight can fail to extract implicit aspects and opinions (Cai et al., 2021; Peper and Wang, 2022). Secondly, during training, the semantic information of the labels has yet to be fully utilized. Traditional supervision utilizes labels in the form of one-hot probability vectors without fully leveraging the semantic information of the labels at the hidden state level.

**TAGS** To address the challenges mentioned

---

*Corresponding author.

above, we propose a novel tagging-assisted generation model called TAGS, which enhances the supervision of both the encoder and decoder through multiple-perspective tagging assistance and label semantic representations. TAGS consists of two modules: "Empowering Generation through Sequence Tagging" (**EGST**) and "Label-Driven Semantic Alignment" (**LDSA**).

In **EGST**, we utilize a sequence tagging task to enhance the generation task through three aspects: Multitask Learning, Guided Generation, and Result Optimization. **Multitask learning**: we enhance the supervision in the encoder of the generation model by introducing a sequence tagging task. This additional task empowers the encoder to distinguish between triplet and irrelevant words effectively, thereby benefiting the generation task. **Guided Generation**: We incorporate the sequence tagging outputs into the decoder's attention mechanism. This encourages the model to focus more on the keywords identified by the sequence tagging task. **Result Optimization**: Finally, during inference, we utilize the sequence tagging results to optimize the generation results, thereby improving the quality of the results.

In **LDSA**, we further enhance the supervision for the decoder's hidden states in the generation model by utilizing the semantic information conveyed by labels. Firstly, we convert label triplets into a natural context, referred to as a label sentence, and input the label sentence into the TAGS model to obtain a more accurate hidden state, which also serves as a semantic label representation. Subsequently, we dynamically align the hidden states of the decoder to the label's semantic representation according to the comparison results between the tokens corresponding to the semantic representation and the ground truth tokens. By this alignment, the model can better capture the semantic information conveyed by the labels, making the generation more in line with the intended label semantics.

Extensive experimental results validate the effectiveness of the TAGS model. In summary, our contributions to this work are threefold:

1. We propose a novel ASTE generation model, which utilizes sequence tagging to assist the generation via enhancing the supervision of the encoder's hidden state and incorporating sequence tagging probabilities and results to improve the generation process.

2. We obtain the semantic representation of la-

bels at the decoder level and achieve semantic alignment of the decoder's hidden state to the labels in the generation model.

3. The experimental results show that our proposed framework significantly outperforms recent SOTA methods.

## 2 Problem statement

The input of the ASTE task is a sentence $\mathbf{X} = \{x_1, x_2, ..., x_n\}$, where each $x_i$ represents a word and $n$ is the maximum length of the sentence. The goal of the ASTE task is to generate a set of sentiment triplets $\mathbf{T} = \{(\mathbf{a}, \mathbf{o}, \mathbf{s})_k\}_{k=1}^{|T|}$, where $|T|$ means the number of triplets in $T$. Each triplet consists of an aspect term ($\mathbf{a}$), an opinion term ($\mathbf{o}$), and the corresponding sentiment polarity ($\mathbf{s}$) ($s \in \{POS, NEU, NEG\}$ ).

Our proposed TAGS is an encoder-decoder model designed for the generation task, in which the input is a natural sentence and the generation target, i.e., the label sentence, is constructed by concatenating triplets from the set T as follows: $Y = "a_1, o_1, s_1; a_2, o_2, s_2; \ldots; a_k, o_k, s_k"$ , where $a_i$, $o_i$, and $s_i$ correspond to the $i$-th triplet $(a, o, s)_i$.

## 3 Methodology

Fig. 1 shows our proposed TAGS method. TAGS comprises two modules, an **E**mpowering **G**eneration through **S**equential **T**agging module (**EGST**) and a **L**abel-**D**riven **S**emantic **A**lignment (**LDSA**) module. **EGST** leverages sequence tagging task to enhance the generation model in three aspects: Multitask Learning, Guided Generation, and Result Optimization. **LDSA** utilizes a label self-decoding process to obtain the semantic representation of labels and aligns the decoder's hidden states to the semantic representation during training, thereby achieving enhanced semantic supervision for the decoder's hidden states.

### 3.1 Empowering Generation through Sequence Tagging

TAGS leverages sequence tagging to enhance the generation task from multiple perspectives, shown in the right part of Fig. 1. Firstly, TAGS employs a sequence tagging task as an additional task to enhance the supervision of the encoder, thereby improving its ability to differentiate between triplet and irrelevant words. By sharing parameters between the sequence tagging model and the generation model, the enhanced discriminative power

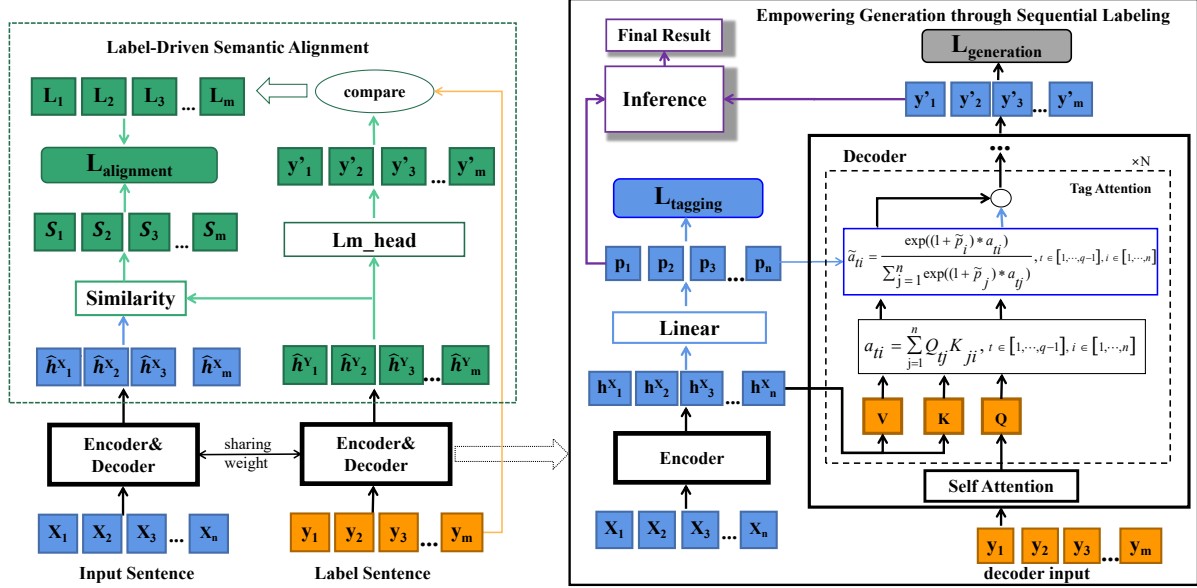

Figure 1: Overview of our TAGS framework, which consists of two parts: Empowering Generation through Sequence Tagging (right) and Label-Driven Semantic Alignment (left). Sequence tagging enhances the generation task in three aspects. **Encoder Multitask Learning**: $\mathcal{L}_{\text{tagging}}$ for sequence tagging task. **Guided Generation**: the decoder's Tag Attention incorporates probabilities ($p_i$) from the sequence tagging task as additional weights for its cross-attention mechanism. (The figure illustrates the generation of the $q$-th word.) **Result Optimization**: "Inference" optimizes the generation results using tagging results. **Semantic representation** of the label ($\hat{h}_{De}^{Y}$) is obtained by inputting the label sentence ($Y$) into the model. $\mathcal{L}_{\text{alignment}}$ is computed based on the cosine similarity results ($\hat{h}_{De}^{X}$ and $\hat{h}_{De}^{Y}$) and the alignment labels $L$.

obtained from the sequence tagging task can also benefit the triplets extraction process in the generation model. Next, the sequence tagging task probabilities are integrated into the generation model, compelling it to prioritize the words identified as crucial by the sequence tagging results. This integration ensures that the generation model produces content closely aligned with those words. Lastly, TAGS utilizes the sequence tagging task results during inference to optimize the generated results. By considering the results from both methods, TAGS achieves a more comprehensive information fusion, enhancing overall model performance.

### 3.1.1 Sequence Tagging Task

We perform multitask learning by simultaneously training a generation task and a sequence tagging task.

**Tagging Scheme** In our designed sequence tagging scheme, each word will be classified into one of 7 categories. The "N" category represents non-keywords, while the remaining 6 categories represent aspects and opinions, each combined with three sentiment types (positive, negative, and neutral). Read Appendix A.1 for detailed descriptions.

**Tagging Task** A tagging sample is de-

noted as $(\mathbf{X}, \mathbf{Z})$, where $\mathbf{Z}$ is the tagging label $\{z_1, z_2, z_3, \ldots, z_n\}$. The encoder encodes $X$ to obtain hidden states $H_{\text{En}}^{X}$:

$$H_{\text{En}}^{X} = En([x_1, x_2, \ldots, x_n]) = [h_1^X, h_2^X, \ldots, h_n^X] \quad (1)$$

where $En$ is Encoder, $H_{\text{En}}^{X} \in \mathbb{R}^{n \times d}$, $d$ denotes the hidden dimension. $H_{\text{En}}^{X}$ is also the encoder hidden state for the generation task. Pass $H_{\text{En}}^{X}$ through a fully connected layer to obtain the tag probabilities $p_i$:

$$p_i = \text{softmax}(W_1 h_i^X + b_1) \quad (2)$$

where $W_1 \in \mathbb{R}^{7 \times d}$, $b_1 \in \mathbb{R}^7$, and $p_i \in \mathbb{R}^7$ represents the probability distribution of the $i$-th word across 7 tags. We calculate the sequence tagging loss using cross-entropy loss:

$$\mathcal{L}_{\text{tagging}} = \sum_{i=1}^{n} \text{CE}(p_i, z_i) \quad (3)$$

### 3.1.2 Generation guided by sequence tagging

**Tag Attention** To leverage the guidance information the sequence tagging task provides, we compute the probability $\widetilde{p}_i$ of the $i$-th word being a keyword. $p_i[0]$ denotes the probability of the $i$-th

word belonging to the non-keyword category ("N"). Consequently, $\widetilde{p}_i = 1 - p_i[0]$ indicates the probability of the $i$-th word belonging to the keyword category. We incorporate $\widetilde{p}_i$ into the cross-attention mechanism of the decoder in the generation model as follows:

$$\widetilde{a}_{ti} = \frac{exp((1 + \widetilde{p}_i) \cdot a_{ti})}{\sum_{j=1}^{n} exp((1 + \widetilde{p}_j) \cdot a_{tj})} \quad (4)$$

where $a_{ti}$ represents the attention score at the $t$-th row and $i$-th column in the cross-attention score matrix before applying softmax. $\widetilde{a}_{ti}$ denotes the final adjusted attention score after applying softmax. $(1 + \widetilde{p}_i)$ ensures a balanced contribution from both the sequence tagging task and inherent generation task to the attention distribution. Compared to the formulation without adding 1 to $\widetilde{p}_i$ (Appendix A.3), this formulation effectively enhances the generation process while mitigating the potential impact of tagging errors on overall generation quality (Appendix B.2).

**Generation Task**  A sample is denoted as $(\mathbf{X}, \mathbf{Y})$, where $\mathbf{Y}$ represents the label sentence $\{y_1, y_2, y_3, \ldots, y_m\}$, with $m$ being the maximum length of Y. The loss function for the generation task with model parameters $\theta$ is defined as follows:

$$\mathcal{L}_{\text{generation}} = -\sum_{t=1}^{m} \log p_\theta(y_t | X, y_{<t}) \quad (5)$$

### 3.1.3 Inference

During inference, we leverage the sequence tagging results to optimize the generated outputs. The main operation involves comparing the generated triplets from the generation task with the triplets from the sequence tagging task. If the generated aspect is a subset of the sequence tagging aspect set, or vice versa, and the generated opinion is a subset of the sequence tagging opinion set, or vice versa, the triplet is retained. Otherwise, the triplet is discarded. Read Appendix A.2 for details.

### 3.2 Label-Driven Semantic Alignment

The form of supervision, similar to Equation 5, lacks fine-grained supervision at the hidden state level and fails to fully utilize the semantic information embedded in the labels. In Label-Driven Semantic Alignment (shown in the left part of Fig. 1), we employ a label self-decoding process to obtain a more accurate decoder hidden state, which serves as a semantic representation of the label. During

training, We align the decoder's hidden state to the semantic representation, thereby enhancing the supervision of the decoder's hidden state. This alignment ensures that the generated output closely matches the semantic content of the label.

**Label Semantic Representation**  During training, we input the label sentence $Y$ into the model to obtain the decoder's hidden state:

$$H_{\text{De}}^Y = En\text{-}De([y_1, y_2, \ldots, y_m]) = [\hat{h}_1^Y, \hat{h}_2^Y, \ldots, \hat{h}_m^Y] \quad (6)$$

where $En\text{-}De$ means encoder-decoder architecture. Since the label sentence contains only the words of the correct triplets, the model can effortlessly extract the correct triplets from it. In this case, the model's input and output are both the label sentence, essentially forming a self-decoding process. Furthermore, due to the absence of irrelevant words in the input, $H_{\text{De}}^Y$ is more accurate compared to $H_{\text{De}}^X$, where $H_{\text{De}}^X = De(H_{\text{En}}^X) = [\hat{h}_1^X, \hat{h}_2^X, \ldots, \hat{h}_m^X]$, as demonstrated in Experiment 4.3.3. Therefore, we regard $H_{De}^Y$ as an accurate semantic representation of the label that can provide substantial supervision at the decoder stage.

**Alignment Labels**  The main objective of semantic alignment is to establish alignment between $H_{\text{De}}^X$ and $H_{\text{De}}^Y$. One significant challenge arises from the fact that even though $H_{\text{De}}^Y$ represents a more accurate hidden state, its corresponding output tokens $Y'$, as shown in Equation 7, may not always match the ground truth token sequence $Y$ during the early stages of training. Therefore, we compare $y_i'$ with $y_i$, and only when $y_i'$ is equal to $y_i$, it indicates that $\hat{h}i^Y$ is correct. We then allow $\hat{h}_i^X$ to be close to $\hat{h}_i^Y$. Otherwise, we move $\hat{h}_i^X$ away from $\hat{h}_i^Y$. Use $L_i$ to represent the comparison result between $y_i'$ and $y_i$:

$$Y' = (\text{Lm\_head}\,(H_{\text{De}}^Y)).\text{argmax}() \quad (7)$$
$$L_i = \text{Equal}(y_i', y_i) \quad (8)$$

where Lm_head represents a linear layer that takes the decoder's hidden states as input and outputs a probability distribution over the vocabulary. The predicted tokens $Y'$ are obtained by selecting the words with the highest probability using the $argmax$ operation. The function "Equal" outputs 1 when the inputs are equal and 0 otherwise.

**Alignment Task**  Alignment is achieved by adjusting the distance between $\hat{h}_i^X$ and $\hat{h}_i^Y$ accoding to $L_i$. Employ cosine similarity to quantify the distance:

$$s_i = \cos(\hat{h}_i^X, \hat{h}_i^Y) \quad (9)$$
$$s_i' = \text{ReLu}(s_i) \quad (10)$$

where cos is the cosine similarity, and ReLu is used to limit the similarity values between 0 and 1 (Appendix B.3). We compute the alignment loss using binary cross-entropy to enforce the cosine similarity scores to align with the labels $L$:

$$\mathcal{L}_{\text{alignment}} = \sum_{i=1}^{m} \text{BCEloss}(s'_i, L_i) \qquad (11)$$

**Final Loss.** Therefore, the final loss is defined as follows:

$$\mathcal{L} = \alpha_1 \mathcal{L}_{\text{generation}} + \alpha_2 \mathcal{L}_{\text{tagging}} + \alpha_3 \mathcal{L}_{\text{alignment}} \qquad (12)$$

where $\alpha_1$, $\alpha_2$ and $\alpha_3$ are hyperparameters.

## 4 Experiments

### 4.1 Experiment Setup

**ASTE Dataset** We evaluate our TAGS on four popular ASTE datasets shown in Table 1: 14Res, 14Lap, 15Res, 16Res (Pontiki et al., 2014, 2015, 2016), which are modified for ASTE task by Fan et al. (2019); Peng et al. (2020); Xu et al. (2020a); Wu et al. (2020).

**Baseline Models** We categorize the comparison models into the following three types:

1.Sequence tagging-based models, such as OTE-MTL (Zhang et al., 2020), GTS (Wu et al., 2020), JET (Xu et al., 2020b), EMC-GCN (Chen et al., 2022a), SyMux (Fei et al., 2022), SCEDD (Zhang et al., 2022b), BDTF (Zhang et al., 2022a), SA-Transformer (Yuan et al., 2023), STAGE (Liang et al., 2023).

2.Generation-based models, such as GAS(Zhang et al., 2020), Paraphrase (Wu et al., 2020), BARTABSA (Yan et al., 2021), PASTE (Mukherjee et al., 2021), Seq2Path (Mao et al., 2022), DLO (Hu et al., 2022a), LEGO-ABSA (Gao et al., 2022), EHG (Lv et al., 2023) and Mvp (Gou et al., 2023).

3.Models based on other methods: reinforcement learning based model ASTE-RL (Jian et al., 2021), reading comprehension based model BMRC (Chen et al., 2021), and span-level models Span-ASTE (Xu et al., 2021) and SBN (Chen et al., 2022b).

**Experiment Details** We employ the T5-base model (Raffel et al., 2020) from the huggingface Transformer library as our pre-trained generative encoder-decoder model. During training, we set the learning rate to $3e$-$4$ for T5 and $5e$-$3$ for all the

Table 1: Statistics of datasets. S and T mean the total number of sentences and triplets. POS, NEU, and NEG represent the number of positive, neutral, and negative sentiment triplets, respectively.

| Dataset | | S | T | POS | NEU | NEG |
|---|---|---|---|---|---|---|
| 15Res | train | 605 | 1013 | 783 | 25 | 205 |
| | dev | 148 | 249 | 185 | 11 | 53 |
| | test | 322 | 485 | 317 | 25 | 143 |
| 16Res | train | 857 | 1394 | 1015 | 50 | 329 |
| | dev | 210 | 339 | 252 | 11 | 76 |
| | test | 326 | 514 | 407 | 29 | 78 |
| 14Lap | train | 906 | 1460 | 817 | 126 | 517 |
| | dev | 219 | 345 | 169 | 36 | 140 |
| | test | 328 | 541 | 364 | 63 | 114 |
| 14Res | train | 1266 | 2337 | 1015 | 50 | 329 |
| | dev | 310 | 577 | 252 | 11 | 76 |
| | test | 492 | 994 | 407 | 29 | 78 |

linear layers. The model is trained for 40 epochs on Nvidia 3090 GPUs, and the hyperparameters of Equation 12 are set as follows: $\alpha_1 = 10$, $\alpha_2 = 1$, and $\alpha_3 = 1$. The probability threshold in the inference stage is 0.999. All the reported results are the average of five runs with different random seeds.

**Evaluation Metrics** Following previous works (Peng et al., 2020), we employ widely used evaluation metrics, namely $F_1$ scores ($F_1$), recall ($R$), and precision ($P$).

### 4.2 Main Results

The main results are reported in Table 2. In this task, F1 is the most important metric (Peng et al., 2020; Chen et al., 2022b; Gao et al., 2022; Gou et al., 2023). TAGS significantly outperforms the previous state-of-the-art method Mvp (Gou et al., 2023), specifically achieving a lead of up to 3.13% on the 16res dataset and 2.01% on the 15res dataset according to the $F_1$ metric.

Based on the principles of sequence tagging-based methods, these approaches tend to be conservative, which means they only predict a triplet when they are highly confident. Consequently, the precision of these methods tends to be higher than the recall, as shown in both the OTE-MTL and JET methods in Table 2. In contrast, generation methods tend to over-predict the number of triplets due to their strong creativity. Consequently, the recall in the results of generation methods is generally higher than the precision.

By introducing a sequence tagging task, the

Table 2: Main results on 4 datasets of ASTE tasks. The best results are in bold, while the second best are underlined. † denotes the replication results, while the other results are obtained from original papers.

| | Model | 16res | | | 15res | | | 14lap | | | 14res | | |
|---|---|---|---|---|---|---|---|---|---|---|---|---|---|
| | | P | R | F1 | P | R | F1 | P | R | F1 | P | R | F1 |
| Other | BMRC | 71.20 | 61.08 | 65.75 | 68.51 | 53.40 | 60.02 | 70.55 | 48.98 | 57.82 | 75.61 | 61.77 | 67.99 |
| | ASTE-RL | 67.21 | 69.69 | 68.41 | 65.45 | 60.29 | 62.72 | 64.80 | 54.99 | 59.50 | 70.60 | 68.65 | 69.71 |
| | Span-ASTE | 69.45 | 71.17 | 70.26 | 62.18 | 64.45 | 63.27 | 63.44 | 55.84 | 59.38 | 72.89 | 70.89 | 71.85 |
| | SBN | 71.59 | 72.57 | 72.08 | 69.93 | 60.41 | 64.82 | 65.68 | 59.88 | 62.65 | 76.36 | 72.43 | 74.34 |
| Tagging | OTE-MTL | 62.88 | 52.10 | 56.96 | 56.37 | 40.94 | 47.13 | 49.53 | 39.22 | 43.42 | 62.00 | 55.97 | 58.71 |
| | JET | 70.42 | 58.37 | 63.83 | 64.45 | 51.96 | 57.53 | 55.39 | 47.33 | 51.04 | 70.56 | 55.94 | 62.40 |
| | GTS | 66.08 | 69.91 | 67.93 | 62.59 | 57.94 | 60.15 | 57.82 | 51.32 | 54.36 | 67.76 | 67.29 | 67.50 |
| | EMC-GAN | 64.43 | 72.63 | 67.69 | 60.45 | 62.72 | 61.55 | 59.61 | 56.30 | 57.90 | 70.37 | 72.84 | 71.58 |
| | SyMux | \ | \ | 72.76 | \ | \ | 63.13 | \ | \ | 60.11 | \ | \ | 74.84 |
| | SCEDD | 66.11 | 71.37 | 68.64 | 59.41 | 62.73 | 61.03 | 61.84 | 60.08 | 60.95 | 70.27 | 73.02 | 71.62 |
| | SA-Transformer | 72.01 | 62.87 | 67.13 | 62.82 | 58.31 | 60.48 | 61.28 | 48.98 | 54.44 | 70.76 | 65.85 | 68.22 |
| | BDTF | 71.44 | 73.13 | 72.27 | 68.76 | 63.71 | 66.12 | 68.94 | 55.97 | 61.74 | 75.53 | 73.24 | 74.35 |
| | STAGE | **77.67** | 68.44 | 72.75 | **72.33** | 58.93 | 64.94 | **70.56** | 55.16 | 61.88 | **78.51** | 69.30 | 73.61 |
| Generation | GAS | \ | \ | 70.10 | \ | \ | 62.10 | \ | \ | 60.78 | \ | \ | 72.16 |
| | Paraphrase | \ | \ | 71.70 | \ | \ | 62.56 | \ | \ | 61.13 | \ | \ | 72.03 |
| | BARTASA | 66.6 | 68.68 | 67.62 | 59.14 | 59.38 | 59.26 | 61.41 | 56.19 | 58.69 | 65.52 | 64.99 | 65.25 |
| | PASTE | 66.1 | 69.8 | 67.9 | 61.7 | 60.8 | 61.3 | 61.2 | 53.6 | 57.1 | 66.7 | 66.5 | 66.6 |
| | DLO | \ | \ | 72.23 | \ | \ | 63.52 | \ | \ | 61.33 | \ | \ | 72.02 |
| | Seq2path† | 71.59 | 75.41 | 73.40 | 62.62 | 65.48 | 64.02 | 64.57 | 60.04 | 62.22 | 73.28 | **74.23** | 73.75 |
| | LEGO-ABSA | \ | \ | 69.9 | \ | \ | 64.4 | \ | \ | 62.2 | \ | \ | 73.7 |
| | EHG | \ | \ | 72.35 | \ | \ | 63.58 | \ | \ | 61.53 | \ | \ | 71.82 |
| | MvP | \ | \ | 73.48 | \ | \ | 65.89 | \ | \ | 63.33 | \ | \ | 74.05 |
| | TAGS | 76.37 | **76.85** | **76.61** | 70.23 | **65.73** | **67.90** | 65.11 | **62.20** | **64.53** | 77.38 | 72.86 | **75.05** |

TAGS method alleviates the excessive creativity of the generation model by directing its focus toward keywords. This not only enhances the quality of the generated output but also objectively limits the number of excessively generated triplets. Leveraging the semantic alignment with labels, TAGS further enhances the quality of the generated triplets. Consequently, compared to conventional generation methods, our method can extract more correct triplets with fewer predicted triplets. This leads to higher precision, recall, and consequently, a higher F1 score. Furthermore, when compared to conventional sequence tagging methods, TAGS surpasses them due to the generation model's ability to utilize semantic information from the labels and its inherent creativity. Thus, TAGS outperforms most previous methods in terms of F1 score, precision, and recall.

Table 3: Ablation study. The results reported are the average F1 scores.

| Model | 16Res | 15Res | 14Lap | 14Res |
|---|---|---|---|---|
| Full Model | 76.61 | 67.90 | 64.53 | 75.05 |
| w/o Tagging traing | 72.83 | 64.34 | 62.15 | 72.96 |
| w/o Tag Attention | 75.68 | 66.51 | 63.14 | 74.18 |
| w/o Inference | 75.49 | 66.66 | 64.19 | 74.34 |
| w/o Alignment | 75.37 | 65.82 | 63.20 | 73.44 |

## 4.3 Ablation

The results of the ablation experiments are presented in Table 3.

**Effectiveness of the Sequence Tagging Task**: The "w/o Tagging training" condition denotes the removal of the sequence tagging task, including multitask training, tag attention, and the specialized inference stage. It means that the model only relies on the Semantic Alignment component. Compared to the "Full Model", the performance under this condition decreased in all datasets: 16res

(-3.78%), 15res (-3.56%), 14Lap (-2.38%), and 14Res (-2.09%), providing evidence for the effectiveness of the sequence tagging task. To further investigate the role of the sequence tagging task, we conducted Experiments 4.3.1.

**Effectiveness of Tag Attention:** The "w/o Tag attention" condition refers to the absence of tag attention while still retaining the training of the sequence tagging task, special inference stage, and the Semantic Alignment component. When compared to the "full model," there was an average performance decrease of 1.11% across all datasets, providing evidence for the effectiveness of Tag Attention. In Appendix B.2, we further analyze the impact of different utilization methods for sequence tagging probabilities on Tag Attention. This analysis enables us to gain a deeper understanding of how the utilization of sequence tagging probabilities influences the performance of Tag Attention.

**Effectiveness of Inference:** The "w/o Inference" condition refers to the absence of a special inference stage. In comparison to the "Full model," there was an average performance decrease of 0.85% across all datasets. This provides evidence for the effectiveness of the Inference stage. In Experiment 4.3.2, we further investigate the experimental results related to the threshold hyperparameter in the inference stage.

**Effectiveness of the Semantic Alignment**: The "w/o Alignment" condition refers to the removal of the Semantic Alignment component. Compared to the "Full Model", the performance under this condition decreased in all datasets: 16res (-1.24%), 15res (-2.08%), 14Lap (-1.33%), and 14Res (-1.61%). This demonstrates the effectiveness of the Semantic Alignment component in improving overall performance. To further investigate the impact of the loss function on the Semantic Alignment component, we conducted Experiment B.3.

### 4.3.1 Loss Hyperparameters

In this section, we investigate the impact of loss hyperparameters. First, we fix $\alpha_2$ and vary $\alpha_1$ and $\alpha_3$, as shown in Fig. 2(a). As $\alpha_1$ gradually increases, the performance initially improves and then decreases, achieving the best result at 10. Comparing the three curves in the graph, the curve corresponding to $\alpha_3 = 1$ achieves the best result. Next, we fix $\alpha_1 = 10$ and vary $\alpha_2$ and $\alpha_3$ as shown in Fig. 2(b). As the $\alpha_2$ increases, the performance initially improves and then decreases, achieving the best result at 1. Furthermore, the curve corresponding to

Table 4: F1 results on the development dataset for different thresholds.

| threshold | 16res | 15res | 14lap | 14res |
|---|---|---|---|---|
| 0.9 | 77.14 | 73.91 | 62.14 | 65.88 |
| 0.99 | 77.62 | 74.27 | 62.61 | 66.00 |
| 0.999 | 77.73 | 74.35 | 62.78 | 66.08 |
| 0.9999 | 77.66 | 74.29 | 62.77 | 66.00 |

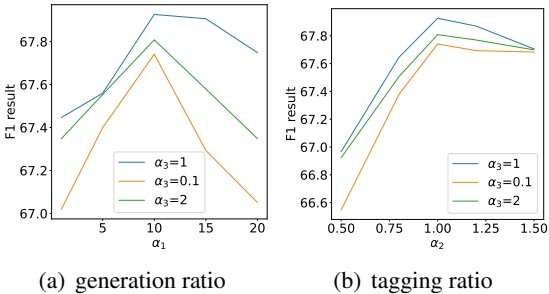

(a) generation ratio     (b) tagging ratio

Figure 2: $F_1$ result with different loss ratios.

$\alpha_3 = 1$ achieves the best result. We select the loss ratios corresponding to the optimal performance as our final hyperparameter settings: $\alpha_1 = 10$, $\alpha_2 = 1$, $\alpha_3 = 1$. This suggests that our method primarily focuses on the generative task, with the other two components serving as auxiliary factors.

### 4.3.2 Threshold Hyperparameter in Inference

We conducted experiments on the development set to determine the most suitable probability threshold hyperparameter. We experimented with four different values for the threshold hyperparameter. The results are shown in Table 4. As the threshold increases, the performance initially improves and then decreases, achieving the best result at 0.999. This threshold value is very close to 1. In the generated results, the probability of each word is also very close to 1, even for some incorrect words. Therefore, when we require a threshold to filter out potentially erroneous triplets, this threshold should also be very close to 1. Hence, 0.999 is a reasonable choice.

### 4.3.3 Correctness of Semantic Representation

To demonstrate that $H_{\text{De}}^Y$ is more accurate, during self-decoding, we replace each label sentence with the original input sentence with a probability of $r$. This increases the influence of irrelevant words on semantic representation. We then train the TAGS model using the semantic representation obtained from this self-decoding process and the correspond-

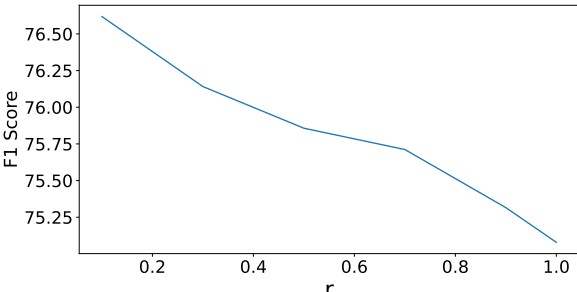

Figure 3: $F_1$ result with different $r$.

ing performance reflects the correctness of the semantic representation. We conducted experiments on the 16res dataset and the results are presented in Fig. 3. The results indicate that as $r$ increases, performance decreases. This demonstrates that an increasing number of irrelevant words in the input lead to a decrease in the correctness of the semantic representation, resulting in a gradual decline in performance.

### 4.4 Results on Other ABSA Tasks

The proposed model provides a unified framework to effectively address the Aspect-Based Sentiment Analysis (ABSA) problem. To demonstrate the effectiveness of TAGS and its generalizability across different tasks, we conducted experiments on two ABSA tasks: AOPE and UABSA. We compared TAGS with the models in Appendix B.4.

AOPE focuses on extracting (aspect, opinion) pairs, similar to ASTE, but without sentiment analysis. This task requires accurate identification of keywords in the sequence tagging task, as well as the assistance of Tag Attention and Semantic Alignment components. The F1 results for the AOPE task are presented in Table 5. TAGS outperforms the previous model on all four datasets: 2.28% for 16Res, 1.50% for 14Lap, 0.83% for 15Res, and 0.51% for 14Res. The improvement in results demonstrates the effectiveness of the aforementioned components.

UABSA focuses on extracting (aspect, sentiment) pairs, similar to ASTE, but without extracting opinions. This task presents challenges in accurately classifying sentiments in sequence tagging and aligning sentiments in Semantic Alignment. The F1 results for the UABSA tasks are presented in Table 6. TAGS has achieved an average improvement of 1.27% compared to the previous model. This improvement demonstrates the effectiveness of the aforementioned components in enhancing

| Model | 16Res | 14Lap | 15Res | 14Res |
|---|---|---|---|---|
| HAST+TOWE(Zhang et al., 2021b) | 63.84 | 53.41 | 58.12 | 62.39 |
| JERE-MHS(Zhang et al., 2021b) | 67.65 | 52.34 | 59.64 | 66.02 |
| SpanMlt(Zhao et al., 2020) | 71.78 | 68.66 | 64.68 | 75.60 |
| SDRN(Chen et al., 2020) | 73.67 | 66.18 | 65.75 | 73.30 |
| GAS(Zhang et al., 2021b) | 74.54 | 68.08 | 67.19 | 74.12 |
| LEGO(Gao et al., 2022) | 77.6 | 69.7 | 71.4 | 78.1 |
| EHG(Lv et al., 2023) | 78.19 | 69.05 | 69.11 | 77.17 |
| **TAGS (Our)** | **80.47** | **71.20** | **72.23** | **78.61** |

Table 5: Main F1 results of the AOPE task. The best results are in bold, second best results are underlined.

| Model | 14Lap | 16Res | 15Res | 14Res |
|---|---|---|---|---|
| BERT+GRU(Li et al., 2019b) | 61.12 | 70.21 | 59.60 | 73.17 |
| SPAN-BERT(Hu et al., 2019) | 61.25 | - | 62.29 | 73.68 |
| MN-BERT (Li et al., 2019b) | 61.73 | - | 60.22 | 70.72 |
| RACL(Chen and Qian, 2020) | 63.40 | - | 66.05 | 75.42 |
| Dual-MRC(Mao et al., 2021) | 65.94 | - | 65.08 | 75.95 |
| GAS(Zhang et al., 2021b) | 67.37 | 71.87 | 65.75 | 75.77 |
| EHG(Lv et al., 2023) | 68.48 | 77.12 | 70.04 | 79.32 |
| **TAGS (Our)** | **71.37** | **78.11** | **70.76** | **79.80** |

Table 6: Main F1 results of the UABSA task. The best results are in bold, second best results are underlined.

the accuracy of sentiment analysis.

These results demonstrate the effectiveness and generalization of TAGS across different tasks.

## 5 Related Work

ASTE employed sequence tagging methods, when it was first introduced by (Peng et al., 2020). Subsequent research efforts (Xu et al., 2020b; Wu et al., 2020; Chen et al., 2022a; Liang et al., 2022; Gou et al., 2023) have been focused on enhancing the sequence tagging schemes and model components to facilitate the integration and mutual interpretation of the triple elements. However, the sequence tagging technique in ASTE fails to capture the semantic information conveyed by the labels, which can lead to semantic mismatches in the predicted results(Zhang et al., 2021b). Generation methods were initially proposed by Zhang et al. (2021c). The generation-based approach in ASTE has achieved good performance by reducing potential error propagation present in pipeline methods and effectively utilizing the rich semantic information provided by labels(Paolini et al., 2021; Yu et al., 2023). They employed various targets for generation, such as sentiment element sequences (Zhang et al., 2021c,c; Hu et al., 2022b), natural language (Liu et al., 2021; Zhang et al., 2021a), and structured extraction patterns (Lu et al., 2022). Recently proposed models, LEGO-ABSA (Gao et al., 2022), UnifiedABSA (Wang et al., 2022) and Mvp (Gou et al., 2023), have focused on leveraging task prompts or guided design for multi-task processing.

## 6 Conclusion

In this work, we introduce a generation model called TAGS, which enhances the supervision of both the encoder and decoder through multiple-perspective tagging assistance and label semantic representations. Specifically, TAGS utilizes sequence tagging to enhance the generation model in multiple aspects: Multitask Learning, Guided Generation, and Result Optimization. Additionally, TAGS employs a label self-decoding process to obtain semantic representations of labels and aligns the decoder's hidden states with these representations, thereby providing enhanced semantic supervision for the decoder's hidden states. These two components enhance the supervision of the encoder and decoder's hidden states, resulting in improved generation quality. Extensive experiments demonstrate that our method significantly advances the state-of-the-art on benchmark datasets.

## 7 Limitations

Despite achieving state-of-the-art performance, our proposed methods still have some limitations that point to potential future directions.

1. Compared to conventional generation methods, our approach requires an additional generation step to obtain more accurate hidden states, namely semantic labels. As a result, there is an increase in training overhead.

2. Although we apply a simple yet effective aggregation strategy to combine the results of the sequence tagging task and generation task, more advanced strategies can be explored to further enhance performance.

3. We have indeed observed that the improvement of our model varies on different datasets, which may be due to the differences in the characteristics of these datasets.

4. Our work utilizes a relatively simple sequence tagging approach, specifically characterized by the absence of explicit pairing between extracted aspects and opinions. There is room for designing a more robust and sophisticated sequence tagging scheme that can also seamlessly integrate with generation models, thereby enhancing performance.

## 8 Ethics Statement

In all our experiments, we used existing datasets that have been widely used in previous scientific publications. When analyzing the experimental results, we strive to maintain fairness and honesty, ensuring that our work does not cause harm to anyone.

As for broader implications, this work may contribute to further research in sentiment analysis and the use of generation methods to simplify and automate the extraction of user opinions in real-world applications. However, it is important to note that this work involves fine-tuning large-scale pre-trained language models to generate sentiment triplets. Due to the nature of the Internet-based large-scale pre-training corpora, the predicted sentiment polarities may be influenced by unintended biases related to gender, race, and intersectional identities (Tan and Celis, 2019). LPMLs often inherit biases present in their training data, potentially leading to biased sentiment analysis results, particularly when assessing text from underrepresented or marginalized groups, thereby perpetuating and amplifying societal prejudices. Another limitation is the opacity of these models. Their complex architectures make it challenging to fully understand the reasoning behind their predictions, raising concerns about transparency and accountability. This lack of interpretability may hinder the identification and mitigation of harmful biases and ethical violations in sentiment analysis applications. It is crucial for the natural language processing community to consider these biases more extensively. Fortunately, these issues are actively being addressed within the research community, including efforts to standardize datasets and methodologies.

## 9 Acknowledgements

We would like to thank the anonymous reviewers for their insightful comments. This work is partially supported by National Natural Science Foundation of China (Grants No. 62176271), and Science and Technology Program of Guangzhou (Grant No. 202201011681).

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

Table 7: Descriptions of the tagging scheme. It focuses on the aspect attribution of each word. The sequence tagging task is to classify each word to one of these tags.

| tag | Meaning |
|---|---|
| N | not belong to aspect term or opinion term |
| A-POS | a part of aspect term with positive sentiment |
| A-NEG | a part of aspect term with negative sentiment |
| A-NEU | a part of aspect term with neural sentiment |
| O-POS | a part of opinion term with positive sentiment |
| O-NEG | a part of opinion term with negative sentiment |
| O-NEU | a part of opinion term with neural sentiment |

learning framework for pair-wise aspect and opinion terms extraction. In *Proceedings of the 58th Annual Meeting of the Association for Computational Linguistics*, pages 3239–3248, Online. Association for Computational Linguistics.

# A   Additional details about the methodology

## A.1   Tagging Scheme

Specific details of the sequence tagging scheme and their explanations are presented in Table 7.

## A.2   Inference

Here's an overview of the inference process:

1. Obtain the probabilities of each word in the generated sentence.

2. Conduct experiments on the development dataset to find a suitable probability threshold.

3. For each triplet in the generated result, check if all the words in the triplet have probabilities greater than the threshold. If so, retain the triplet. Otherwise, proceed to the next step.

4. Compare the generated triplet with the triplet identified by the sequence tagging task. If the generated aspect is a subset of the sequence tagging aspect set, or vice versa, and the generated opinion is a subset of the sequence tagging opinion set, or vice versa, retain the triplet. Otherwise, discard the triplet.

The specific algorithm pseudocode is presented in algorithm 1.

## A.3   Other Attention

$$\widetilde{a}_{ti} = \frac{exp(a_{ti})}{\sum_{j=1}^{n} exp(a_{tj})} \tag{13}$$

$$\widetilde{a}_{ti} = \frac{exp(\widetilde{p}_i \cdot a_{ti})}{\sum_{j=1}^{n} exp(\widetilde{p}_j \cdot a_{tj})} \tag{14}$$

# B   Additional Experiment

## B.1   case study

In this case study section, we compare our model with the Paraphrase model (Zhang et al., 2021a) to illustrate how our two components benefit the results. For the first example, the Paraphrase model fails to extract the triplet $(barmenu, Disappointingly, NEG)$ because it is a relatively hard and implicit triplet. Additionally, thanks to the semantic alignment training of the decoder hidden state, our model can generate higher-quality results. Therefore, our model can extract this triplet successfully. In the second example, the Paraphrase model incorrectly extracts the triplet $(staff, supportive, NEG)$. However, during the inference stage, our model optimizes the generation results based on the sequence tagging output, resulting in the discarding of this incorrect triplet.

## B.2   Arithmetic Operations in Tag Attention

In the context of Tag Attention, we have explored several approaches to incorporating tagging probability into the cross-attention mechanism:

1. Multiplication before softmax: Multiply the attention scores by the probability weights and then apply softmax.

2. Multiplication is performed before softmax, but without adding 1 to $\widetilde{p}_i$. The attention formula is given by Equation (14).

3. Softmax after multiplication: Apply softmax to the attention scores and then multiply them by the probability weights.

4. Addition: We directly add the probability information to the attention scores.

Through the evaluation of these various operations, our objective is to gain insights into their impact on the Tag Attention mechanism and their effectiveness in incorporating probability information. We conducted this experiment on a model without the "Inference" process because including

Table 8: Case study. The ground truth represents the correct triplets. The aspect and opinion words of the same triplet are highlighted in the same color. The two examples in the table demonstrate how our model can avoid the errors made by the Paraphrase model.

| Example | Ground Truth | Paraphrase (Zhang et al., 2021a) | Ours |
|---|---|---|---|
| **Disappointingly**, their **wonderful Saketini** has been taken off the **bar menu** . | (**Saketini**,**wonderful**,POS) (**bar menu**,**Disappointingly**,NEG) | (**Saketini**,**wonderful**,POS) | (**Saketini**,**wonderful**,POS) (**bar menu**,**Disappointingly**,NEG) |
| Our **waiter** was **friendly** and it is a shame that he didnt have a supportive staff to work with. | (**waiter**,**friendly**,POS) | (**waiter**,**friendly**,POS) (**staff**,**supportive**,NEG) | (**waiter**,**friendly**,POS) |

Table 9: Result of different arithmetic operations

| | 16res | 15res | 14lap | 14res |
|---|---|---|---|---|
| Multiplication before softmax | 76.61 | 67.90 | 64.53 | 75.05 |
| Multiplication before softmax without adding 1 to $\widetilde{p}$ | 75.81 | 66.71 | 63.23 | 74.09 |
| Softmax after multiplication | 73.92 | 65.05 | 61.78 | 72.38 |
| Addition | 73.06 | 64.58 | 61.43 | 72.12 |

Table 10: Result of different loss function

| | 16res | 15res | 14lap | 14res |
|---|---|---|---|---|
| MarginRankLoss | 75.55 | 67.12 | 64.19 | 73.91 |
| BCELoss | 76.61 | 67.90 | 64.53 | 75.05 |

the "Inference" process could potentially narrow down the performance gaps observed in these results. The results in Table 9 demonstrate that the first approach performs better. This is because it provides valuable information to the generation model while minimizing any disruptive effects on the original generation process. It can be regarded as a gentle process. The results of the second approach are worse compared to the first approach. One possible reason for this is that in the first approach, by adding 1 to the $\widetilde{p}$, the attention is not solely determined by the sequence tagging results. This helps mitigate the potential impact of tagging errors on the overall generation quality. Furthermore, we found that the performance of the last two arithmetic operations is worse.

## B.3  Loss Function for Semantic Alignment

We discuss loss function for semantic alignment in our approach. Specifically, we compare two different approaches:

1. Confining the similarity scores to the range of 0 to 1 and utilizing the Binary Cross Entropy (BCE) loss function.

2. Preserving the cosine similarity scores in the range of -1 to 1 and employing the margin ranking loss function to constrain the similarity.

The results in Table 10 indicate that in our method, the BCE loss function outperforms the margin rank loss function. From this, we can conclude that it is not necessary to push the similarity of incorrect hidden states to -1, i.e., there is

no need to excessively move away from the negative hidden states associated with incorrect words. Since the hidden states are generated from the label sentences, even if some negative hidden states are incorrect, they remain relatively close to the correct hidden states. Moving too far away from negative hidden states may lead to an increase in the distance from the correct hidden state.

## B.4  ABSA subtask Detail

The subtasks are described as follows:

1. Aspect Opinion Pair Extraction (AOPE) aims to extract aspect terms and their corresponding opinion terms as pairs (Zhang and Qian, 2020; Chen et al., 2020).

2. Unified ABSA (UABSA) is the task of extracting aspect terms and predicting their sentiment polarities at the same time (Li et al., 2019a; Chen and Qian, 2020). We also formulate it as an (aspect, sentiment polarity) pair extraction problem

For these tasks, we adopt the dataset used in (Zhang et al., 2021b).

For AOPE task, we compare our model with the following models: a multi-task learning model SpanMlt (Zhao et al., 2020), a synchronous double channel extraction model SDRN (Chen et al., 2020), HAST+TOWE and JERE-MHS model compared in (Zhang et al., 2021b), GAS (Zhang et al., 2021b), LEGO(Gao et al., 2022) and EHG(Lv et al., 2023).

For the UABSA task, we compare our model with the following models: a BERT base model BERT+GRU (Li et al., 2019b), a span-base extraction model SPAN-BERT (Hu et al., 2019), an interactive multi-task learning network LMN-BERT

(He et al., 2019), a Relation-Aware Collaborative Learning (RACL) model RACL (Chen and Qian, 2020), a machine reading comprehension models Dual-MRC (Mao et al., 2021) , GAS (Zhang et al., 2021b) and EHG(Lv et al., 2023).

### B.5 Analysis on Potential Practical Applications

Time Complexity: The time complexity of the TAGS model is quadratic relative to the input data. The primary source of complexity in this quadratic time complexity is the attention operations within the transformer. It's important to note that the additional modules introduced in our model, such as the sequence tagging classification layer and the label-driven semantic alignment module, have a linear time complexity relative to the input data. Consequently, the time complexity introduced by our additional modules remains exceedingly modest compared to that of the transformer. As such, the primary temporal overhead in our model stems from the transformer's attention operations. Consequently, the complexity of the TAGS model closely aligns with the time complexity of baseline models that rely on transformers. Moreover, existing lightweight and acceleration-oriented designs based on the transformer can be readily assimilated into our model. Hence, although our model does introduce some additional time overhead, it does not impose a significant obstacle to the training process.

Space Complexity: Apart from the core model architecture and input data, the additional space utilization of the TAGS model primarily consists of a linear layer for sequence tagging classification and the semantic representation of label sentences. The additional space occupation amounts to 5.2 M, which is notably minor when compared to the parameter size of the T5 model, standing at 222 M. Additionally, the tag attention module does not introduce any additional parameters.

Based on the aforementioned explanation, it's evident that the TAGS model demonstrates commendable scalability. As dataset volumes increase, the incremental rise in both time and space overheads within our model remains consistent.

**Input:** Generated triplets $T_1' = \{(a_i, o_i, s_i)_k\}_{k=1}^{|T_1'|}$,
Word probabilities $G=\{g_k = (g_{k1}, g_{k2}, g_{k3})_k\}_{k=1}^{|T_1'|}$,
Tagging Aspect Set $S_{aspect} = \{a_k\}_{k=1}^{|S_{aspect}|}$,
Tagging Opinion Set $S_{opinion} = \{a_k\}_{k=1}^{|S_{opinion}|}$,
Threshold $threshold$.
**Output:** Result triplets

**Function** `Verify`(*gen_element, sequenceTag*)**:**
    **foreach** *tag_element in sequenceTag* **do**
        **if** *gen_element is a part of tag_element OR tag_element is a part of gen_element* **then**
            **return** *true*
        **end**
    **end**
    **return** *false*
**begin**
    Result $\leftarrow \emptyset$
    **foreach** $(triplet, wordProb)$ in $(T_1', G)$ **do**
        **if** *(wordProb $\geq threshold$).all()* **then**
            Result$\leftarrow$ Result $\cup$ {triplet} **end**
        **else**
            aspect $\leftarrow$triplet.aspect
            Averified$\leftarrow$`Verify`(aspect,$S_{aspect}$)
            opinion $\leftarrow$triplet.opinion
            Overified$\leftarrow$`Verify`(opinion,$S_{opinion}$)
            **if** *Averified AND Overified* **then**
                Result$\leftarrow$ Result $\cup$ {triplet}
            **end**
        **end**
    **end**
    **return** Result

**Algorithm 1:** Inference