# OpenReview forum: "Tagging-Assisted Generation Model with Encoder and Decoder Supervision for Aspect Sentiment Triplet Extraction"
_EMNLP/2023/Conference — EMNLP 2023 Main_

### Official Review · Reviewer_Uz23 · 2023-07-29

**Soundness:** 4

**Excitement:**

4: Strong: This paper deepens the understanding of some phenomenon or lowers the barriers to an existing research direction.

**Missing References:**

Following work is also based on encoder-decoder framework and should be cited and compared as a baseline.
[1] PASTE: A Tagging-Free Decoding Framework Using Pointer Networks for Aspect Sentiment Triplet Extraction

**Paper Topic And Main Contributions:**

This paper proposed a tagging-assisted encoder-decoder model for aspect-opinion-sentiment triplet extraction (ASTE). They add an additional sequence labelling task in an multi-task approach and use the label-sequence to improve the decoding process of the model. The sequence labelling component only finds the aspect terms and opinion terms along with the sentiment information. It does not pair the aspect terms with the opinion terms. The decoder module does the job of pairing them with the help of the sequence labels. They carry out experiments with four popular datasets in this task and achieved SOTA performance in four datasets. They also show the effectiveness of their model in various variation task such as AOPE, ABSA.

**Reasons To Accept:**

1. Authors proposed an effective combination of sequence labelling approach and generative approach to improve the performance of the generative models in ASTE task.
2. Experimental results in four datasets show the effectiveness of their proposed approach on ASTE task, as well as on the ABSA and AOPE sub-tasks.
3. Paper is well written and easy to follow. Authors have provided proper ablation studies and analysis to understand the proposed model.

**Reasons To Reject:**

1. To some extent this is an incremental work that uses previously known algorithms and combine them to improve performance on the ASTE tasks.

**Reproducibility:**

4: Could mostly reproduce the results, but there may be some variation because of sample variance or minor variations in their interpretation of the protocol or method.

**Reviewer Confidence:**

4: Quite sure. I tried to check the important points carefully. It's unlikely, though conceivable, that I missed something that should affect my ratings.

---

> ### Author Rebuttal · Authors · 2023-08-29
>
> We would like to express our sincere gratitude to you for your valuable feedback and insights into our work.
>
> **For the reason to reject:**
>
> We genuinely appreciate the reviewer's careful assessment of our work and would like to address the concern that our paper may be perceived as incremental. To clearly illustrate that our contribution extends beyond incremental work, we want to emphasize the following three key points:
>
> **1、Innovative Label Semantics Supervision:**
>
> In our second contribution, we present a novel and creative attempt to enhance the supervision of the decoder in the generation model by utilizing the semantic representation of the labels. Specifically, we perform self-decoding on label sentences, where the decoder's output serves as the semantic representation of labels. This semantic representation is then used to supervise the content generated by the generation model. In essence, we parameterize label sentences to provide fine-grained, semantic-level labels. This is a shift from traditional one-hot label supervision to semantic label supervision. This approach brings forth various advantages, encompassing not only the benefits mentioned in the paper but also extending to capabilities such as disambiguating polysemous words—an intricate task that is not easily tackled through conventional one-hot labels. To the best of our knowledge, ours is the first work to adopt such a strategy. We believe that this novel approach holds significant promise and warrants attention.
>
> **2、The idea of choosing Sequence Tagging:**
> After successfully enhancing supervision over the decoder's hidden states with the semantic representation of labels, we realized that introducing an additional task to enhance supervision over the encoder's hidden states was also imperative. Generally, classification tasks put a strong supervision on the encoder. We believe that a classification-based sequence tagging task better strengthens the encoder's ability to distinguish between triplet and non-triplet elements. So, we have decided to include a sequence tagging task as an additional task to enhance supervision over the encoder.
>
> Furthermore, the sequence tagging task within our model is relatively straightforward. This deliberate design objectively demonstrates that the improvements in our model are not contingent upon the advantages brought by advanced sequence tagging methods.
>
> **3、Meticulously Designed Technical Details:**
>
> We have innovatively and meticulously designed our work, rather than merely assembling existing works.
>
> a) Firstly, to ensure that the generation task remains the central focus, we devised a streamlined sequence tagging scheme. Within this sequence tagging scheme and its associated loss functions, we deliberately omitted the explicit pairing of aspects and opinions. Consequently, this deliberate omission represents a relatively loose constraint. The rationale behind this choice lies in the potential adverse effects of imposing overly stringent constraints on the encoder, which could impede the learning process and the ultimate performance of the generation task. Striking an appropriate balance in constraint strength on the encoder allows us to enhance the encoder's discriminative capacity in identifying elements of the triplets, all the while benefiting the generation process.
>
> b) Furthermore, the design of this sequence tagging scheme enables the output results to be readily utilized by the tag attention component to guide the generation process.
>
> In light of these contributions and distinctions from previously known algorithms, we hope that the reviewer will consider our work as an innovative and valuable contribution to the field of ASTE. We are fully committed to refining our manuscript to better communicate these aspects and to provide a compelling rationale for the innovative nature of our approach.
>
> **For the Missing References:**
>
> We greatly appreciate your suggestion to include the work 'PASTE: A Tagging-Free Decoding Framework Using Pointer Networks for Aspect Sentiment Triplet Extraction' as a baseline in our paper. We acknowledge the significance of conducting comprehensive comparisons with existing methods to ensure a thorough evaluation of our contributions.
>
> After a meticulous review, we have identified that our work indeed shares some similarities with the 'PASTE' framework, as both are built upon the encoder-decoder architecture.
>
> In our revised manuscript, we will incorporate "PASTE" as one of the key baseline methods, along with proper citations. The following Table 1 presents a comparison of our model's F1 performance with that of PASTE. The F1 performance of PASTE is copied from its respective publication.
>
> Table 1. Comparison with the missing reference.
> | Dataset    | 16res | 15res | 14lap | 14res |
> |------------|-------|-------|-------|-------|
> | PASTE      | 67.9  | 61.6  | 57.4  | 66.6  |
> | TAGS(ours) | **76.61** | **67.90** | **64.53** | **75.05** |
>
>
> The TAGS model outperforms the PASTE model (e.g., over 8% improvement in the case of 16res). This disparity in performance can be attributed to the divergent objectives of the two models. While the PASTE model is primarily focused on predicting the position pointers of triple elements, TAGS, on the other hand, is designed to predict the natural language textual forms of the triplets. The TAGS approach, a generation-based method, is capable of leveraging the semantic information of the labels more comprehensively, thus yielding superior results. Furthermore, the TAGS model incorporates a label-driven semantic alignment module, which enables a more profound utilization of the semantic information encoded within the labels.
>
> Finally, it is important to note that the supplementary comparisons we have provided will not impact the experimental conclusions in our manuscript.
>
> This addition will allow us to provide a more comprehensive and detailed comparative analysis, further enhancing the robustness and credibility of our research. We genuinely appreciate your valuable suggestion, and we are committed to ensuring that our revised paper reflects a comprehensive evaluation of our approach with relevant prior work.

---

### Official Review · Reviewer_3TTV · 2023-08-05

**Soundness:** 4

**Excitement:**

4: Strong: This paper deepens the understanding of some phenomenon or lowers the barriers to an existing research direction.

**Paper Topic And Main Contributions:**

This article proposes a new approach called Tagging-Assisted Generation Model with Encoder and Decoder Supervision (TAGS) for Aspect Sentiment Triplet Extraction (ASTE). The proposed model enhances the supervision of encoder and decoder hidden representations in Natural Language Generation-based methods for ASTE by integrating sequence tagging probabilities and semantic representations of labels. The model achieves enhanced semantic information utilization and improves the extraction of implicit aspects and opinions. The main contribution of this work is the proposed TAGS model, which outperforms the state-of-the-art methods on two benchmark datasets for ASTE. The work may contribute to further research in sentiment analysis and the use of generation methods to simplify and automate the extraction of user opinions in real-world applications.

**Questions For The Authors:**

 1.What potential biases may affect the predicted sentiment polarities in this work?
 2.Does the article mention any limitations or challenges of the sequence tagging approach used in this work?
 3.Does the article discuss any potential ethical concerns related to the use of large-scale pre-trained language models in sentiment analysis?

**Reasons To Accept:**

1.The paper highlights the importance of considering biases in large-scale pre-trained language models and the need for more extensive consideration of biases in the NLP community.
2.The creative aspects of this article include proposing a new approach to enhance the supervision of encoder and decoder hidden representations in Natural Language Generation-based methods for ASTE, integrating sequence tagging probabilities and semantic representations of labels
3.The proposed Tagging-Assisted Generation Model with Encoder and Decoder Supervision (TAGS) enhances the supervision of encoder and decoder hidden representations in Natural Language Generation-based methods for Aspect Sentiment Triplet Extraction (ASTE), achieving enhanced semantic information utilization and improving the extraction of implicit aspects and opinions.

**Reasons To Reject:**

1.The paper does not provide a detailed discussion of the potential practical applications of the proposed TAGS model, such as its scalability or its compatibility with different types of data.
2. The paper does not compare the proposed TAGS model with a wider range of state-of-the-art methods for ASTE, which could provide a more comprehensive evaluation of its effectiveness.
3.The paper does not provide a detailed analysis of the limitations of the proposed TAGS model, such as its sensitivity to hyperparameters or its performance on datasets with different characteristics.

**Reproducibility:**

3: Could reproduce the results with some difficulty. The settings of parameters are underspecified or subjectively determined; the training/evaluation data are not widely available.

**Reviewer Confidence:**

3: Pretty sure, but there's a chance I missed something. Although I have a good feel for this area in general, I did not carefully check the paper's details, e.g., the math, experimental design, or novelty.

---

> ### Author Rebuttal · Authors · 2023-08-29
>
> We would like to express our sincere gratitude to you for your valuable feedback and insights into our work.
>
> **For the reason to reject#1 "The paper does not provide a detailed discussion of the potential practical applications of the proposed TAGS model, such as its scalability or its compatibility with different types of data.":**
>
> We appreciate the reviewer's comment regarding the need for a detailed discussion of the potential practical applications, such as scalability, and compatibility of the proposed TAGS model. **In fact, we have already voiced concerns about the additional time overhead associated with the process of obtaining semantic representation of labels in our "Limitations" section: "Compared to conventional generation methods, our approach requires an additional generation step to obtain more accurate hidden states, namely semantic labels. As a result, there is an increase in training overhead." However, it is worth emphasizing that these supplementary time costs do not significantly impede the model's training process.**
>
> In our revised manuscript, we will provide the following comprehensive discussions of these critical aspects.
>
> **Scalability Analysis:**
>
> **Time Complexity:** The time complexity of the TAGS model is quadratic relative to the input data. The primary source of complexity in this quadratic time complexity is the attention operations within the transformer. It's important to note that the additional modules introduced in our model, such as the sequence tagging classification layer and the label-driven semantic alignment module, have a linear time complexity relative to the input data. Consequently, the time complexity introduced by our additional modules remains exceedingly modest compared to that of the transformer. As such, the primary temporal overhead in our model stems from the transformer's attention operations. Consequently, the complexity of the TAGS model closely aligns with the time complexity of baseline models that rely on transformers. Moreover, existing lightweight and acceleration-oriented designs based on the transformer can be readily assimilated into our model. Hence, although our model does introduce some additional time overhead, it does not impose a significant obstacle to the training process.
>
> **Space Complexity:** Apart from the core model architecture and input data, the additional space utilization of the TAGS model primarily consists of a linear layer for sequence tagging classification and the semantic representation of label sentences. The additional space occupation amounts to 5.2 M, which is notably minor when compared to the parameter size of the T5 model, standing at 222 M. Additionally, the tag attention module does not introduce any additional parameters.
>
> Based on the aforementioned explanation, it's evident that the TAGS model demonstrates commendable scalability. As dataset volumes increase, the incremental rise in both time and space overheads within our model remains consistent and manageable.
>
> During practical training processes on datasets of varying scales, our model consistently maintains its performance without any noticeable degradation or efficiency concerns. In terms of computational resources, our TAGS model showcases a modest utilization of GPU memory. For instance, when training on the largest dataset among the four datasets used in our paper, utilizing a 24GB Nvidia 3090 GPU, we can set a batch size of 32. This emphasizes the minimal GPU memory demand of our model. Notably, for the largest dataset among the four, our TAGS model completes training an epoch in less than two minutes. This demonstrates the model's efficiency in terms of training time. Consequently, our TAGS model is well-equipped to support the training of large-scale datasets.
>
> **Compatibility with Different Types of Data:**
>
> The datasets employed in our study, which are also widely recognized datasets, encompass three restaurant review datasets and one laptop review dataset. Sentiment analysis can be applied to a broader range of textual forms, including news commentary and various other domains. Our model exhibits adaptability to a wide spectrum of domains, given that the design of our TAGS model is domain-agnostic and does not rely on domain-specific prior knowledge.
>
> However, existing datasets within the field are predominantly centered around product reviews and lack diversity in terms of text forms, such as news commentary or other textual styles. As a result, conducting direct evaluations on these unexplored data types becomes unfeasible. Nevertheless, we undertook supplementary experiments on datasets from a recent ACL 2023 paper (released in May) to investigate TAGS' compatibility with a broader range of data types. (In adherence to the guidelines, I refrain from providing a direct link during rebuttal.)
>
> This new dataset encompasses social media text and product reviews, distinguishing itself from the datasets used in our paper in several aspects:
>
> **Longer Texts:** The text in this new dataset is notably longer, with an average length approximately 3.6 times that of the commonly used datasets. **Longer sequence lengths in the dataset also facilitate the assessment of our model's scalability when presented with lengthier input data.**
>
> **Diverse Expressions:** The vocabulary within this new dataset is three times larger than that of the commonly used datasets, implying greater diversity and complexity. Moreover, it features more intricate grammar.
>
> **Varied Domains:** The dataset encompasses a multitude of domains, including electronics, beauty, fashion, and home, highlighting its compatibility with a wider array of domains.
>
> We have conducted performance evaluations of the TAGS model on this new dataset and compared it with several models, with the F1 results obtained from the dataset's associated paper.
>
> Table 1. Evaluation of a new dataset from a recent ACL 2023 paper.
> | Dataset    | Electronics | Beauty | Fashion | Home  |
> |------------|-------------|--------|---------|-------|
> | BMRC       | 41.95       | 38.57  | 44.87   | 41.18 |
> | BART-ABSA  | 43.38       | 41.13  | 43.89   | 40.56 |
> | GAS        | 47.10       | 44.32  | 47.80   | 47.22 |
> | TAGS(ours) | **49.97**       | **47.12**  | **50.66**   | **50.24** |
>
> Our results significantly outperform the baseline comparisons, demonstrating improvements of 2.87% (Electronics), 2.8% (Beauty), 2.86% (Fashion), and 3.02% (Home) compared to the GAS model. This underscores the effectiveness of our contributions across multiple domains.
>
> In summary, we will extensively expound upon the potential practical applications, such as scalability, and compatibility of the TAGS model in our revised manuscript. We firmly believe that these insights will fortify the comprehensiveness of our work.
>
> **For the reason to reject#2 "The paper does not compare the proposed TAGS model with a wider range of state-of-the-art methods for ASTE, which could provide a more comprehensive evaluation of its effectiveness.":**
>
> **We have indeed conducted comprehensive comparisons with various state-of-the-art (SOTA) methods, encompassing generation models, sequence labeling models, span-level models, and those based on reading comprehension. Our objective has been to provide an extensive evaluation that encompasses a diverse range of competing methodologies.**
>
> In the revised version of our paper, we are committed to significantly expanding the comparative assessment section. We have diligently sought a broader array of comparative methods within the ASTE domain to ensure a more thorough evaluation of the TAGS model. For instance, we will introduce two additional comparative methods in our study, namely "STAGE: Span Tagging and Greedy Inference Scheme for Aspect Sentiment Triplet Extraction" and "Boundary-Driven Table-Filling for Aspect Sentiment Triplet Extraction". The following table presents a comparison of our model's F1 performance with that of the two papers. These results have been excerpted from their respective publications.
>
> Table 2. Comparison with two additional SOTA methods.
> |   Dataset  | 16res | 15res | 14lap | 14res |
> |:----------:|:-----:|:-----:|:-----:|:-----:|
> |    STAGE | 72.75 | 64.79 | 61.88 | 73.61 |
> |    BDTF | 72.27 | 66.12 | 61.74 | 74.35 |
> | TAGS(ours) | **76.61** | **67.90** | **64.53** | **75.05** |
>
> From the tabulated results, it becomes evident that our approach outperforms both of these newly introduced methods. Our model demonstrates superiority over these grid-based sequence labeling models, attributed not only to the nature of our generation task but also to the efficacy of our label-driven semantic alignment module. Our TAGS model notably outperforms the newly introduced comparative methods, providing further confirmation of the strengths of our paper and the effectiveness of our approach.
>
> Finally, it is important to note that the supplementary comparisons we have provided will not impact the experimental conclusions in our manuscript.
>
> By incorporating these supplementary methods, we now present a more robust comparison of TAGS against a diverse set of baselines, showcasing its performance superiority and underscoring its unique contributions. We anticipate that these additional comparisons will enhance the overall quality of our work and its contributions to the ASTE research community.
>
> **For the reason to reject#3 "The paper does not provide a detailed analysis of the limitations of the proposed TAGS model, such as its sensitivity to hyperparameters or its performance on datasets with different characteristics.":**
>
> We sincerely appreciate the reviewer's feedback and recognize the significance of providing a comprehensive analysis of the limitations of our proposed TAGS model. **Within the Limitations section of our paper, we have indeed outlined several limitations of our model, including concerns related to additional time overhead and the simplicity of the sequence tagging scheme.**
>
> In our revised manuscript, we are committed to addressing this concern by offering a more thorough examination of the model's limitations.
>
> **Sensitivity to Hyperparameters:**
>
> Our primary hyperparameter configuration is defined as the generation loss ratio: tag loss ratio: align loss ratio = 10:1:1. Given that our task places a predominant emphasis on the generation task, the generation loss ratio is set relatively high. Our hyperparameter experiments systematically varied the generation loss ratio from 1 to 20, revealing that the optimal performance is achieved at a ratio of 10. In contrast, the other loss components serve as auxiliary constraints and are maintained at values near 1. Further hyperparameter experiments indicate that the ratios near 1 consistently yield the best results, and the results exhibit stable performance within the vicinity of 1.
>
> Crucially, our model's performance is not sensitive to hyperparameters. We have consistently employed the same set of hyperparameters across all four datasets and have consistently outperformed comparative methods using these configurations. The hyperparameter experiments detailed in the paper further validate the effectiveness of these chosen hyperparameters.
>
> **Performance on Datasets with Different Characteristics:**
>
> To begin, it's important to highlight that the TAGS model exhibits significant improvements across all datasets, demonstrating its broad adaptability.
>
> However, it's worth noting that the extent of improvement with our TAGS model is more pronounced on more complex datasets. The simpler datasets are already well-handled by existing models. Due to their straightforward grammatical structures, these datasets pose fewer challenges in terms of erroneous triplet extractions, resulting in higher accuracy for existing models. Additionally, the simpler grammatical structures make correct triplet extractions relatively straightforward, leading to higher recall for existing models. In cases where the grammatical structure is simpler, such as the 14res dataset, our TAGS model doesn't show substantial improvement, as other models perform sufficiently well.
>
> In contrast, our model bolsters the encoder's discriminative capabilities concerning triplet elements. Thus, even in complex datasets, the TAGS model enhances the extraction of implicit triplets, leading to improved recall. The tag attention mechanism guides the model to focus on keywords identified by the sequence tagging task, enabling it to avoid erroneous extractions of ambiguous triplets even in complex datasets. This mechanism enhances accuracy. Therefore, our TAGS model exhibits more substantial improvements on datasets with complex syntax, such as the 16res dataset.
>
> Furthermore, as detailed in our responses above (response to reject reason #1), we have conducted experiments on an additional dataset, specifically the 2023 ACL-released dataset. These datasets, generally more complex than the ones we currently use, provide a valuable testing ground for our TAGS model. Experimental results indicate that our TAGS model achieves significant and consistent improvements, with F1 scores increasing by 2.87% (Electronics), 2.8% (Beauty), 2.86% (Fashion), and 3.02% (Home) compared to the GAS model.
>
> **For question#1 "What potential biases may affect the predicted sentiment polarities in this work? ":**
>
> First and foremost, it is imperative to emphasize that the design of our TAGS model intentionally avoids the introduction of any prior knowledge, aiming to minimize biases. Our model operates on the principles of natural language processing and machine learning and, fundamentally, does not favor any specific sentiment polarity.
>
> However, within the context of our work, it is worth acknowledging that pre-trained models may be exposed to biases in the pre-training data they utilize, potentially leading to biases in the predicted sentiment polarities. This is a challenge that most researchers encounter when utilizing pre-trained models. It is important to note that the issue of biases in pre-trained models has garnered significant attention. Notably, there has been a concerted effort towards curating more standardized training data for pretraining models, thereby mitigating the prevalence of biases. In our study, we employ the pre-trained generative model T5, developed by Google, which utilizes meticulously curated and cleaned datasets. This significantly reduces the potential bias inherited from the pre-training data.
>
> Understanding and rectifying biases is a critical undertaking in the domains of sentiment analysis and natural language processing. We are committed to promoting responsible AI practices and advancing the development of sentiment analysis systems that are more equitable and just.
>
> **For question#2 "Does the article mention any limitations or challenges of the sequence tagging approach used in this work?":**
>
> Indeed, it is noteworthy that the current sequence tagging approaches for the ASTE exhibit certain limitations, and the generation-based model we propose aims to address these limitations to a certain extent. **In our "Limitations" section, we acknowledge that our sequence tagging approach employs a simplified design, specifically characterized by the absence of explicit pairing between extracted aspects and opinions.** However, this limitation does not significantly impact the overall performance, as during the generation phase, the decoder leverages contextual information and comprehensively considers both the aspects and opinions outcomes from the sequence tagging task to combine and pair crucial elements within the sentences.
>
> This deliberate simplification is a strategic choice made for the sequence tagging approach, which is designed for the generation model, and it is motivated by the following reasons:
>
> 1) Firstly, to ensure that the generation task remains the central focus, we devised a streamlined sequence tagging scheme. Within this sequence tagging scheme and its associated loss functions, we deliberately omitted the explicit pairing of aspects and opinions. Consequently, this deliberate omission represents a relatively loose constraint. The rationale behind this choice lies in the potential adverse effects of imposing overly stringent constraints on the encoder, which could impede the learning process and the ultimate performance of the generation task. Striking an appropriate balance in constraint strength on the encoder allows us to enhance the encoder's discriminative capacity in identifying elements of the triplets, all the while benefiting the generation process.
>
> 2) Furthermore, the design of this sequence tagging scheme enables the output results to be readily utilized by the tag attention component to guide the generation process.
>
> **For question#3 "Does the article discuss any potential ethical concerns related to the use of large-scale pre-trained language models in sentiment analysis?":**
>
> In fact, our article does mention a range of ethical concerns associated with the use of large-scale pre-trained language models (LPMLs) in the broader domain of sentiment analysis. **In our Ethics Statement section, we explicitly state: "Due to the nature of the Internet-based large-scale pre-training corpora, the predicted sentiment polarities may be influenced by unintended biases related to gender, race, and intersectional identities."** This statement underscores a critical issue, wherein LPMLs often inherit biases present in their training data, potentially leading to biased sentiment analysis results, particularly when assessing text from underrepresented or marginalized groups.
>
> Furthermore, in the revised version of our paper, we plan to delve more deeply into ethical considerations. Fortunately, these issues are actively being addressed within the research community, including efforts to standardize datasets and methodologies.
>
> Specifically, within the scope of our task, as we primarily focus on product reviews, ethical considerations may not be as frequently encountered. Nonetheless, we remain vigilant about the broader ethical implications associated with LPMLs and sentiment analysis, and we are committed to contributing to responsible AI practices.

---

### Official Review · Reviewer_Gsym · 2023-08-05

**Soundness:** 4

**Excitement:**

4: Strong: This paper deepens the understanding of some phenomenon or lowers the barriers to an existing research direction.

**Missing References:**

More ASTE tagging methods should be included for comparison.
1. https://www.ijcai.org/proceedings/2022/572
2. https://ieeexplore.ieee.org/document/9857593
3. https://ieeexplore.ieee.org/document/10175600

**Paper Topic And Main Contributions:**

This paper presents a tagging-assisted generation model with encoder and decoder supervision (TAGS) for the ASTE task. The contribution is a new ASTE generation model with the SOTA results.

**Questions For The Authors:**

Why Chen et al., 2022b is not included for comparison?

**Reasons To Accept:**

This paper presents a new ASTE generation model and outperforms the SOTA methods.

**Reasons To Reject:**

None

**Reproducibility:**

4: Could mostly reproduce the results, but there may be some variation because of sample variance or minor variations in their interpretation of the protocol or method.

**Reviewer Confidence:**

4: Quite sure. I tried to check the important points carefully. It's unlikely, though conceivable, that I missed something that should affect my ratings.

---

> ### Author Rebuttal · Authors · 2023-08-29
>
> We would like to express our sincere gratitude to you for your valuable feedback and insights into our work.
>
> **For question #1 "Why Chen et al., 2022b is not included for comparison?":**
>
> We apologize for not comparing with the work by Chen et al., 2022b in our paper. It was an oversight on our part. The results of the work by Chen et al., 2022b are listed in Table 1 and discussed in the following paragraphs. We will include the comparison with Chen et al., 2022b in the revised version of our paper.
>
> **For the "Missing References"**:
>
> We appreciate your suggestion to include more ASTE tagging methods for comparison. We have listed the results of these methods in Table 1 and discussed them in the following paragraphs. In our revised paper, we will incorporate a comparative analysis of our TAGS model and these additional methods, along with proper citations. This will ensure a more comprehensive evaluation of the TAGS model's performance in ASTE tagging methods.
>
> Table 1 provides a concise comparison of the F1 performance results of our TAGS model and the works the reviewer referred to. The F1 performance is extracted from the respective publications of Chen et al., 2022b, and the three papers mentioned by the reviewers for comparison.
>
> Table 1. Comparison with Chen et al., 2022b and other ASTE tagging methods
> |       Dataset      | 16res | 15res | 14lap | 14res |
> |:------------------:|:-----:|:-----:|:-----:|:-----:|
> |    Chen et al., 2022b | 72.08 | 64.82 | 62.65 | 74.34 |
> |        the first paper | 72.76 | 63.13 | 60.11 | 74.84 |
> |  the second paper    | 68.64 | 61.54 | 60.95 | 71.62 |
> |   the third paper    | 67.13 | 60.48 | 54.44 | 68.22 |
> |     TAGS(ours)     | **76.61** | **67.90** | **64.53** | **75.05** |
>
> The results table illustrates that the TAGS model outperforms the results achieved by Chen et al. (2022b) to a significant degree (e.g., over 4% improvement in the case of 16res). This is largely attributed to the TAGS model's superior ability, in comparison to Chen's model, to comprehensively consider contextual information when pairing aspect and opinion elements.
>
> Furthermore, the TAGS model outperforms the three sequence tagging models. For instance, compared to the first paper (the best one) of the three sequence tagging methods paper, about 4% improvements are achieved in the former three datasets. Our model demonstrates superiority over these sequence tagging models, attributed to the advantage of our generation task, which is assisted by tagging and aligned by the label-driven semantics.
>
> Finally, it is important to note that the supplementary comparisons we have provided will not impact the experimental conclusions in our manuscript.

---

### Meta-Review · Area_Chair_R5ms · 2023-09-08

**Recommendation:** 5

**Metareview:**

This paper proposes a new method for aspect sentiment triplet extraction. The authors propose a new approach to enhance the supervision of encoder and decoder's hidden representations in Natural Language Generation-based methods for ASTE, integrating sequence tagging probabilities and semantic representations of labels. This leads to new SOTA results. The minor points for improvement include more discussion of the application possibilities of the proposed methods, as well as a comparison to more prior methods, both of which have been provided in the author response.

---

### Decision · Program_Chairs · 2023-10-07

**Decision:**

Accept-Main

**Comment:**

This paper proposes a new method for aspect sentiment triplet extraction. The authors propose a new approach to enhance the supervision of encoder and decoder's hidden representations in Natural Language Generation-based methods for ASTE, integrating sequence tagging probabilities and semantic representations of labels. This leads to new SOTA results. The minor points for improvement include more discussion of the application possibilities of the proposed methods, as well as a comparison to more prior methods, both of which have been provided in the author response.